



# On the effectiveness of recession analysis methods for capturing the characteristic storage-discharge relation: An intercomparison study

Xing Chen [1,2], Mukesh Kumar [1], Stefano Basso [3], Marco Marani [1,4]

[1]Nicholas School of the Environment, Duke University, Durham, 27708, USA
[2]Rays Computational Intelligence Lab, Beijing Inteliway Environmental Ltd, Beijing, 100871, China
[3]Department of Catchment Hydrology, Helmholtz Centre for Environmental Research, Halle, 06120, Germany
[4]Department ICEA, University of Padova, Padova, 35122, Italy

*Correspondence to*: Mukesh Kumar (mukesh.kumar@duke.edu)

**Abstract.** Storage-discharge ($S$-$Q$) relations are widely used to derive watershed properties and predict streamflow responses. These relations are often obtained using different recession analysis methods, which vary in recession period identification criteria and $Q$ vs. $-dQ/dt$ fitting scheme. Although previous studies have indicated that different recession analysis methods can result in significantly different $S$-$Q$ relations, several challenges remain regarding the evaluation of

relative effectiveness of these methods in obtaining the characteristic $S$-$Q$ relation. Here we demonstrated these challenges and presented a new "control setup" based experimental approach to compare four recession analysis methods. Results indicated that irregular binning and event-based methods show superior performance at obtaining the characteristic $S$-$Q$ relation and reconstructing streamflow, while lower envelope method performs the worst. Notably, accuracy of the methods is influenced by the extent of scatter in the $\ln(-dQ/dt)$ vs. $\ln(Q)$ plot. In addition, the derived $S$-$Q$ relation was very sensitive

to the criteria used for identifying recession periods. These results raise a warning sign against indiscriminate application of recession analysis methods and derived $S$-$Q$ relations for watershed characterizations or hydrologic simulations. Thorough evaluation of representativeness of the derived $S$-$Q$ relation should be performed before it is used for hydrologic analysis.



## 1 Introduction

Storage-discharge (*S-Q*) relations are used to understand and predict hydrologic responses [*Bart and Hope*, 2014; *Biswal and Marani*, 2010; *Ceola et al.*, 2010; *Harman and Sivapalan*, 2009; *Kirchner*, 2009; *Palmroth et al.*, 2010; *Rupp and Selker*, 2006b; *Teuling et al.*, 2010; *Wang*, 2011] and to characterize watershed properties such as drainable porosity, soil

conductivity and aquifer thickness [*Brutsaert and Nieber*, 1977; *Brutsaert and Lopez*, 1998; *Stoelzle et al.*, 2013; *Szilagyi*, 2003; *Szilagyi et al.*, 2007]. These relations are often derived by identifying a relation between stream discharge, $Q$, and its time derivative, $-dQ/dt$, during recession periods of the hydrograph when evapotranspiration and rapid flow contributions (e.g., surface and subsurface flows) to the discharge are negligible, and streamflow is primarily determined by the catchment storage [*Brutsaert and Nieber*, 1977]. The relation between $-dQ/dt$ and $Q$ has been derived using a multitude of recession

analysis methods [*Basso et al.*, 2015; *Biswal and Marani*, 2010; *Brutsaert and Nieber*, 1977; *Kirchner*, 2009; *Shaw and Riha*, 2012; *Stoelzle et al.*, 2013; *Szilagyi et al.*, 2007; *Teuling et al.*, 2010; *Vogel and Kroll*, 1992], which primarily differ in the procedures used to:

(a) *identify streamflow recession periods when rapid flow and evapotranspiration contributions to the streamflow are minimal*. The time it takes for groundwater flow to be dominant in the streamflow after a precipitation event depends on

watershed properties. Hence, it is challenging to select a time interval beyond which contributions of rapid flow and evapotranspiration to the recession hydrograph are negligible, particularly when such an analysis is done based on climate and streamflow data alone. Previous studies have used thresholds ranging from one to 10 days after rainfall events [*Bart and Hope*, 2014; *Brutsaert and Nieber*, 1977; *Brutsaert and Lopez*, 1998; *Malvicini et al.*, 2005; *Mendoza et al.*, 2003; *Rupp et al.*, 2004; *Szilagyi and Parlange*, 1998; *Troch et al.*, 1993; *Van Dijk*, 2010; *Vogel and Kroll*, 1992; *Zecharias and Brutsaert*,

1988].

(b) *fit a regression line through the scatter points in* $\ln(-dQ/dt)$ *vs.* $\ln(Q)$ *plot.* For example, *Brutsaert and Nieber* [1977], *Rupp et al.* [2009] and *Palmroth et al.* [2010] fitted a straight line through the lower envelope of the scatter plot, as the lowest $-dQ/dt$ for a given $Q$ is likely determined only by groundwater flow with minimal influence from overland flow, evapotranspiration, interflow or channel storage. In contrast, *Vogel and Kroll* [1992], *Kirchner* [2009], *Ceola et al.* [2010],

*Teuling et al.* [2010], *Ajami et al.* [2011], *Staudinger et al.* [2011], and *Stoelzle et al.* [2013] fitted a regression line through all the scatter points. *Biswal and Marani* [2010], *Shaw and Riha* [2012], *Mutzner et al.*, [2013], *Shaw et al.* [2013], *Bart and Hope* [2014], *Biswal and Marani* [2014], *Biswal and Kumar* [2014], and *Basso et al.* [2015] on the other hand derived a separate regression line for each recession event using only the points from that event.

Different recession analysis methods yield different *S-Q* relations, consequently affecting derived hydrologic

variables [*Basso et al.*, 2015; *Ceola et al.*, 2010; *Chen and Krajewski*, 2016; *Dralle et al.*, 2017; *Stoelzle et al.*, 2013]. Even when using the same method, the derived hydrologic variables may still vary to a great extent with small changes in aforementioned procedures [*Szilagyi et al.*, 2007]. However, this has largely been overlooked in majority of previous studies where a single recession analysis method was generally used to derive the *S-Q* relation and the derived relation was used *as*





*is* for subsequent hydrologic analysis without any evaluation of its validity or representativeness. The problem is partly attributable to the lack of fine temporal resolution (e.g., daily) watershed-wide storage data that can be used to directly validate the derived *S-Q* relation.

5       To circumvent this problem, some studies used an indirect technique to compare the performances of different methods. This indirect technique involved using the derived *S-Q* relation within a modeling framework to generate hydrologic variables that may be validated against the observed data. For example, *Ceola et al.* [2010] used four different methods within an analytical model proposed by *Botter et al.* [2009] to generate probability density functions (PDFs) of streamflow and compared the simulated PDFs against those obtained from observations. Their results showed that a binning method, similar to the one proposed by *Kirchner* [2009], provided the best parameters which described *S-Q* relations in five

out of 14 considered watersheds. Using the same analytical model, *Basso et al.* [2015] compared three different methods for estimating the cumulative probability of daily high flows, and suggested that $-dQ/dt$ and $Q$ relation obtained from individual recession events outperformed the other two methods in 31 out of 43 cases.

      While both *Ceola et al.* [2010] and *Basso et al.* [2015] compared the ability of recession analysis methods to estimate PDFs and flow duration curves (FDCs), their studies evaluated the derived *S-Q* relations and the analytical

framework in unison. It is not clear if the comparison between methods will still be valid when the assumptions used in the analytical model are no longer valid. For instance, the analytical model proposed by *Botter et al.* [2009] relied on the assumption that subsurface/groundwater flow is the dominant contribution to streamflow and its suitability is limited to watersheds with "absence of extensive impermeable surfaces". However, as shown in *Chen et al.* [2015], annual surface flow contributions to streamflow can exceed 40%, even when less than 10% of the watershed area is urban. Also, the two

studies did not address the influence of procedure used to identify the recession period on representativeness of derived *S-Q* relations. Furthermore, their focus was limited to comparing different methods' ability to generate streamflow PDFs. It is still not known how the choice of a recession analysis method may influence the accuracy with which streamflow time series are estimated.

      Building on the aforementioned inter-comparison studies, here we assessed the performance of four recession

analysis methods for deriving the *S-Q* relation. Specifically, we assessed the influence of the methodology used to identify recession periods dominated by groundwater flow and to fit a regression line through the $\ln(-dQ/dt)$ *vs.* $\ln(Q)$ scatter on the accuracy of the derived *S-Q* relations. Accuracy of these derived *S-Q* relations was evaluated through direct comparison of parameters in the *S-Q* relation, provided the characteristic *S-Q* relation of the watershed was known. Otherwise, accuracy was evaluated based on the subsequently reconstructed streamflow time series and its PDFs. The comparisons were

performed over 45 watersheds, using two years long meteorological and streamflow data. Analysis periods were chosen to ensure identical data length across the study sites. Considering the pace of urban land cover expansion in southeastern U.S [*Homer et al.*, 2015], the two years period also indirectly guaranteed that watershed properties such as land use and land cover would have not changed appreciably over the analysis period.





The study is organized as follows. Section 2 presents the descriptions of the study sites, workflow of the four recession analysis methods and the strategy we used to evaluate methods' performance. Section 3 describes the performance of each method using both observed and synthetic data. Section 4 summarizes the results and takeaways from this study.

## 2 Methodology

We employed four methods with different strategies to identify groundwater flow dominated recession periods and to fit regression lines: (1) "Lower envelope method (LEM)" by *Brutsaert and Nieber* [1977], (2) "Central tendency method (CTM)" by *Vogel and Kroll* [1992], (3) "Irregular binning method (IBM)" by *Kirchner* [2009] and (4) "Event-based method (EBM)" by *Basso et al.* [2015]. The chosen methods broadly span the majority of recession analysis methodologies used in literature. Details of the study areas are presented in Section 2.1. Criteria used to identify groundwater flow dominated
recession periods and strategies to fit regression lines to $\ln(-dQ/dt)$ vs. $\ln(Q)$ scatterplots are outlined in Section 2.2. Methodologies for evaluating the accuracy of the derived *S-Q* relations are presented in Section 2.3, whereas Section 2.4 describes the metrics used to evaluate methods' performances.

### 2.1 Study sites and data

Forty-five rain-dominated watersheds (Table 1) from seven southeastern U.S. states were selected for analysis. All the
watersheds have drainage areas less than 25 km$^2$ and are drained by perennial streams. Climatology in all 45 watersheds is humid [*Kottek et al.*, 2006] (average annual precipitation ranging from 641 mm to 1787 mm and runoff ratio ranging from 0.16 to 0.96). Daily streamflow data are available from the U.S. Geological Survey for the period 2011-2015. The fraction of missing values in the streamflow data for each year was less than 1%. Climate data (e.g., hourly precipitation, air temperature, solar radiation, wind speed, and relative humidity) for these watersheds were obtained from Phase 2 dataset of
the North American Land Data Assimilation System (NLDAS-2) [*Xia et al.*, 2012]. Potential evapotranspiration (*PET*) was calculated based on FAO-Penman-Monteith equation [*Allen et al.*, 1998].

### 2.2 Recession analysis methods

Recession analysis methods aim to estimate the relation between catchment water storage *S* and river discharge *Q* based on recession flow data. Considering mass conservation, the water budget equation at the basin scale can be written as:

$$\frac{dS}{dt} = P - ET - Q \tag{1}$$

where *P* is precipitation and *ET* is evapotranspiration. Assuming $g(Q) = dQ/dS$ as a sensitivity function which describes the magnitude of change in discharge per unit change in storage, Eq. (1) can be re-written as





$$\frac{dS}{dt} = \frac{dS}{dQ}\frac{dQ}{dt} = \frac{1}{g(Q)}\frac{dQ}{dt} = P - ET - Q \ \ \text{or} \ \ \frac{dQ}{dt} = g(Q)(P - ET - Q) \tag{2}$$

Considering periods with negligible $P$ and $ET$, the equation reduces to

$$-\frac{dQ}{dt} = g(Q) \cdot Q \tag{3}$$

For watersheds where $S$-$Q$ relation follows a power-law form, i.e., $S = \alpha Q^{\beta}$, the equation can be re-written in logarithmic

scale as

$$\ln\left(-\frac{dQ}{dt}\right) = -\ln(\alpha\beta) + (2 - \beta)\ln(Q) \tag{4}$$

This indicates that a plot between $\ln(-dQ/dt)$ vs. $\ln(Q)$, for periods with negligible $P$ and $ET$, can be used to estimate $\alpha$ and $\beta$ parameters. The following subsections detail the procedures used for estimating coefficients $\alpha$ and $\beta$ in the four selected recession analysis methods. $-dQ/dt$ and $Q$ in all four methods were obtained using $-dQ/dt = (Q_t - Q_{t-1})/\Delta t$ and $Q = (Q_t + Q_{t-1})/2$

respectively, where $\Delta t$ is a constant value ($\Delta t = 1$ day) unless stated otherwise. The calculations were all performed at daily time interval, same as the temporal resolution of the streamflow data.

### 2.2.1 Lower envelop method (*LEM*)

This method was first proposed by *Brutsaert and Nieber* [1977] and later adopted and modified by *Brutsaert and Lopez* [1998], *Rupp and Selker* [2006b], *Brutsaert and Sugita* [2008], *Palmroth et al.* [2010], *Wang and Cai* [2010] and *Wang*

[2011]. Streamflow data beginning from 5 days after any rainfall event were recorded as a recession period. The relation between $S$ and $Q$ was then obtained by fitting a least-squares regression line through the lower envelope of the $\ln(-dQ/dt)$ vs. $\ln(Q)$ scatter. When the magnitude of streamflow change is smaller than the precision of the stream gauge, the derived lower envelope of $\ln(-dQ/dt)$ vs. $\ln(Q)$ could be an artifact [*Rupp and Selker*, 2006a]. Therefore, we followed *Palmroth et al.* [2010] and adopted a "scaled-$\Delta t$" method to obtain $-dQ/dt$, where instead of using a constant $\Delta t$, a varying $\Delta t$ was used.

$\Delta t$ was calculated at each observation point based on the following criteria: if $Q_{t+\Delta t} - Q_{t-\Delta t} \geq 0.001 \times \overline{Q}$ for $\Delta t = 1$ day, then $\Delta t$ was kept at the current value, otherwise, $\Delta t$ was gradually increased until $Q_{t+\Delta t} - Q_{t-\Delta t} \geq 0.001 \times \overline{Q}$. Here $\overline{Q}$ was the average streamflow magnitude of the entire streamflow time series. Once all $-dQ/dt$ and $Q$ points for the recession period were obtained and plotted in the log-log axis, the lower envelope of the recession plot was identified through the following process based on *Palmroth et al.* [2010]: data were divided into six $\ln(Q)$ bins and a "low value" of $\ln(-dQ/dt)$ was

calculated for each bin. The "low value" was the mean $\ln(-dQ/dt)$ of all points that fall below $c$ times the standard deviation from the mean of that bin. Although $c = 2$ was used by *Palmroth et al.* [2010], we found that the lower envelope method performed better for $c = 0.5$ for the selected watersheds (see Fig. S1 and Table S1 in Supplementary Materials). Hence, $c =$



0.5 was selected for analysis in this study. The "low values" of ln($-dQ/dt$) for the six bins were then regressed against the mean of ln($Q$) in each bin using linear least-squares fitting to obtain $\alpha$ and $\beta$.

### 2.2.2 Central tendency method (*CTM*)

This method was first used in *Vogel and Kroll* [1992] to estimate regional low-flow statistics and later used in *Ceola et al.*
[2010]. Recession periods dominated by groundwater flow were identified as periods starting from the time points when a 3-day moving average of $Q$ begins to decrease and ending when it begins to increase. In the original method, only recession periods that were longer than 10 days and with $Q_t \geq 0.7Q_{t-1}$ at each time step of the recession were selected. However, due to the high frequency of precipitation in the southeastern U.S., the minimum length of 10 days for recession periods excluded almost all recession events in the selected watersheds. Hence, we modified the criterion and considered all recession periods with duration of three or more days. Periods with streamflow smaller than the 5th percentile were excluded from the analysis, in order to remove points possibly affected by measurement error. A similar exclusion was performed in *Vogel and Kroll* [1992] and *Palmroth et al.* [2010]. The parameters $\alpha$ and $\beta$ were then obtained by fitting a line through all the ln($-dQ/dt$) vs. ln($Q$) data points using a linear least-squares method.

### 2.2.3 Irregular binning method (*IBM*)

This method was first developed by *Kirchner* [2009] and subsequently used in a number of studies including *Ceola et al.*
[2010], *Teuling et al.* [2010], *Ajami et al.* [2011] and *Staudinger et al.* [2011]. Here, recession periods dominated by groundwater flow were identified as intervals when precipitation and evapotranspiration fluxes were small compared to discharge. In *Kirchner* [2009], recession periods were identified using hourly records. Since here we worked with daily data, recession periods were assumed to start two days after a rainfall event following *Basso et al.* [2015]. Only periods with potential evapotranspiration less than 25th percentile were selected. ln($-dQ/dt$) vs. ln($Q$) data points were first divided into 100 bins equally spaced along the ln($Q$) axis. Starting from this minimum bin size and the maximum value of $Q$, the bin size was increased locally and additional data points were included until the standard error in $-dQ/dt$ became lower or equal to the mean of $-dQ/dt$ in that bin. Once the bins were determined, the average values of $-dQ/dt$ and $Q$ for each bin were calculated. Using linear least-squares regression weighted by the reciprocal of the square of the standard errors of that bin, $\alpha$ and $\beta$ parameters were obtained through fitting ln($-dQ/dt$) vs. ln($Q$) data. This approach weighs high $Q$ values more and limits the influence of low $Q$ values on the regression, as low $Q$ values are more likely to have measurement errors caused by instruments' precision and stage-discharge relations [*Rupp and Selker*, 2006a].

### 2.2.4 Event-based method (*EBM*)

The event-based method used here is based on the implementation in *Basso et al.* [2015]. Here, recession periods were identified as intervals beginning two days after precipitation events until the start of the next storm and with continuous





decrease in streamflow for at least five days. A linear least-squares regression was then fitted through $(\ln(-dQ/dt), \ln(Q))$ data pairs for each individual recession event to estimate $\beta_e$. $\beta$ was estimated as the median of $\beta_e$ values from all the recession events. By keeping $\beta_e$ constant and equal to $\beta$, $\alpha_e$ was then estimated using linear least-squares regression for each recession event. $\alpha$ was estimated by evaluating the median of the re-calculated $\alpha_e$ for all the recession events [*Bart and*

*Hope*, 2014; *Basso et al.*, 2015].

The recession periods and regression lines derived using the four aforementioned methods are illustrated in Fig. 1 for watershed 9 (Table 1).

### 2.3 Evaluating recession analysis methods

The four recession analysis methods were evaluated based on their capability to capture the characteristic *S-Q* relation. In

this regard, $g(Q)$ was derived from the recession hydrograph using Eqs. (3) and (4), which can be rearranged as

$$\ln(g(Q)) = \ln\left(\frac{-dQ/dt}{Q}\right) = c_1 + c_2 \ln(Q) \tag{5}$$

where $c_1 = -\ln(\alpha\beta)$ and $c_2 = 1-\beta$. Note that $c_1$ and $c_2+1$ are intercept and slope of the regression line fitted through $-dQ/dt$ vs. $Q$ scatter in log-log scale, respectively. The derived $g(Q)$ from different recession analysis methods could then be compared with the "true" value to evaluate the accuracy of each method. However, the "true" $g(Q)$ or the "true" *S-Q* relation

is generally unknown for real watersheds due to lack of daily watershed-wide storage data. Hence, we performed an indirect evaluation based on the accuracy of reconstructed streamflow using the derived *S-Q* relation (Fig. 2a). Streamflow reconstruction was performed using the strategy detailed in *Kirchner* [2009], based on which Eq. (2) was rewritten as

$$\frac{d(\ln(Q))}{dt} = g(Q)\left(\frac{P-ET}{Q}-1\right) \tag{6}$$

Reconstructed streamflow was then obtained using the following equation

$$\ln(Q)_{t+\Delta t} = \ln(Q)_t + g(Q_t)\left(\frac{P_t - ET_t}{Q_t}-1\right)\Delta t \tag{7}$$

where $ET$ was computed as $ET=\lambda \cdot PET$ and $\lambda$ was a constant evaluated based on $\lambda=(P-Q)/PET$ for the entire simulation period. As the change in $Q$ may lag behind the change in $P$ and $S$ due to infiltration and transport processes in the subsurface, we calculated the cross correlation between $P$ and $dQ/dt$ to identify the lag time [*Kirchner*, 2009]. This lag time was then applied to $P$ and $ET$ time series when using Eq. (7). Readers are referred to paragraphs 40 and 41 of *Kirchner* [2009] to

know more about the details of the lagging procedure. We used ode15s function in Matlab™ 2015b to solve the first order differential equation in Eq. (7). It is to be noted that reconstructed streamflow time series obtained using Eq. (7) is expected to be inaccurate (/accurate) if $g(Q)$ estimated from a recession analysis method based on Eq. (5) has errors (/is error free). An accurate reconstructed streamflow would indicate that $g(Q)$ function, and hence the *S-Q* relation, was accurately captured.





The streamflow reconstruction strategy presented above was based on the assumption that streamflow depends solely on catchment storage, and catchment storage can be represented as a single storage element [*Kirchner*, 2009]. However, these assumptions may not be valid in the studied watersheds. In addition, the accuracy of reconstructed streamflows may be biased by errors in measurements of meteorological forcings and streamflows, errors introduced by

mapping inputs across scales, and by the approximation involved in the calculation of *ET*. To circumvent the challenges posed by these assumptions and approximations, recession analysis methods were also evaluated using a "control experiment" setup (Fig. 2b). Here, synthetic streamflow time series ($Q_{syn}$) was generated for each watershed based on the following equation

$$\ln(Q_{syn})_{t+\Delta t} = \ln(Q_{syn})_t + g_{syn}(Q_{syn}) \left( \frac{P_t - ET_t}{(Q_{syn})_t} - 1 \right) \Delta t \tag{8}$$

where $P$, $ET$ and $g_{syn}(Q_{syn})$ were predefined. Theoretically, any $P$, $ET$ and $g_{syn}(Q_{syn})$ may be used. To make $Q_{syn}$ more realistic, $g_{syn}(Q_{syn})$ was set equal to the derived $g(Q)$ from observation data, i.e., $g_{syn}(Q_{syn})=g(Q)$. Meanwhile, observed $P$ data from each watershed was used, while $ET$ time series was set equal to $\lambda \cdot PET$. Since the observed $\ln(-dQ/dt)$ vs. $\ln(Q)$ plot always shows a scatter or spread around the "true" $g(Q)$ (e.g., Fig. 1), we also considered a $g_\varepsilon(Q_{syn\_\varepsilon})$ function with a Gaussian white noise, i.e., $g_\varepsilon(Q_{syn\_\varepsilon})=g_{syn}(Q_{syn})+\varepsilon$. Chi-square goodness-of-fit tests at 5% significance level confirmed that $\varepsilon$

exhibited a normalized Gaussian distribution in 37 out of 45 watersheds. Magnitude of this Gaussian white noise qualitatively captured the scatter or the magnitude of uncertainties in the $\ln(-dQ/dt)$ vs. $\ln(Q)$ plot (shown in row 3 of Fig. 1). Two levels of noise magnitudes were considered in the $g_\varepsilon(Q_{syn\_\varepsilon})$ function: level 1 noise ($\varepsilon 1$) had the same mean but half the standard deviation of the observed, while level 2 noise ($\varepsilon 2$) had the same mean and standard deviation as that in the observation data. By individually substituting $g_{syn}(Q_{syn})$ with $g_\varepsilon(Q_{syn\_\varepsilon 1})$ and $g_\varepsilon(Q_{syn\_\varepsilon 2})$ in Eq. (8), we generated synthetic

streamflows $Q_{syn\_\varepsilon 1}$ and $Q_{syn\_\varepsilon 2}$, respectively. Recession analysis methods were applied to the two synthetic streamflow time series to extract sensitivity functions $g(Q_{syn\_\varepsilon 1})$ and $g(Q_{syn\_\varepsilon 2})$. Performance of each method was then evaluated using both direct and indirect approaches (Fig. 2). In the direct approach, methods were evaluated based on their capability to extract the "true" sensitivity function. In other words, $g(Q_{syn\_\varepsilon 1})$ and $g(Q_{syn\_\varepsilon 2})$ derived from each recession analysis method were compared with $g_{syn}(Q_{syn})$. In the indirect approach, performance of each method was evaluated based on how well the

reconstructed streamflow, obtained by individually substituting $g(Q)$ with $g(Q_{syn\_\varepsilon 1})$ and $g(Q_{syn\_\varepsilon 2})$ in Eq. (7), compared with $Q_{syn}$.

### 2.4 Quantifying and comparing the performances of recession analysis methods

In the indirect approach for evaluating different recession analysis methods, we quantified the accuracy of four reconstructed variables, namely streamflow time series, its PDF for the entire length of the series, and the high flow and low flow PDFs.

High flow was defined as the streamflow magnitude higher than the 80$^{th}$ percentile, while low flow was defined as the





streamflow lower than the 20th percentile. A weighted coefficient of determination, $wr^2$, metric was used to quantify difference between variables derived from the base and reconstructed series. Observed streamflow was the base streamflow series for real watersheds (Fig. 2a), whereas $Q_{syn}$ was the base streamflow for analyses on synthetic streamflow series (Fig. 2b). $wr^2$ was chosen as it can quantify both the dispersion and systematical bias of the reconstructed variables. In contrast, $r^2$

ignores the latter information, i.e., if the reconstructed variables were systematically over- or under- estimated, $r^2$ would still be high [*Krause et al.*, 2005]. $wr^2$ was calculated based on:

$$wr^2 = r^2 \begin{cases} |b| \, (|b| \le 1) \\ |b|^{-1} \, (|b| > 1) \end{cases}$$   (9)

where

$$r^2 = \frac{\sum_{i=1}^{n} (B_i - \overline{B})(R_i - \overline{R})}{\sqrt{\sum_{i=1}^{n} (B_i - \overline{B})^2} \sqrt{\sum_{i=1}^{n} (R_i - \overline{R})^2}}$$   (10)

$B$ and $R$ were the base and reconstructed streamflows, $b$ was the slope parameter of the linear least-squares fitted regression between $B$ and $R$. $wr^2$ nearer to 1 indicated a good fit. For each watershed, $wr^2$ was evaluated for each method and for all four reconstructed variables. $wr^2$ evaluation for high and low flow PDFs involved identifying periods in the base series during which the streamflow magnitude was over 80th or under 20th percentile, respectively. For these identified high and low flow periods, we generated corresponding PDFs with 100 equal sized bins for both the base and reconstructed series.

The range of $Q$ in both base and reconstructed PDFs were set identical, with minimum and maximum magnitude encompassing the full range of $Q$ among the two series. $wr^2$ performance was then evaluated based on comparisons between the base and reconstructed PDFs for the identified periods. An average value of $wr^2$ among all watersheds was then calculated for each method. Since average $wr^2$ of a method may get biased by its exceptionally good/bad performance in just a few watersheds, we also evaluated the performance of each method by computing total ranks of $wr^2$ for the different

reconstructed variables (Table 2). In this regard, $wr^2$ was sorted in descending order to rank the methods. For example, to identify the best performing method in terms of its ability to reconstruct the streamflow, the method that had the largest $wr^2$ was ranked 1 and the one with the lowest $wr^2$ was ranked 4 for a given watershed. Total rank or the sum of rankings of each method from all the watersheds was then calculated, and the method with the least total rank was regarded as the best method for that reconstruction variable. To validate the robustness of analysis, in addition to $wr^2$, other metrics such as the

Nash-Sutcliffe efficiency (NSE), the logarithm Nash-Sutcliffe efficiency (logNSE), the Root Mean Square Error (RMSE), and the coefficient of determination ($R^2$) were also evaluated (see Table S2 in Supplementary Materials).

For synthetic streamflow, we also performed direct evaluation of the accuracy of derived *S-Q* relations. This was accomplished through the comparison of intercept and slope of the derived sensitivity functions $g(Q_{syn\_\varepsilon1})$ and $g(Q_{syn\_\varepsilon2})$ with that of $g_{syn}(Q_{syn})$ (Fig. 2b).



## 3 Results and discussions

### 3.1 Evaluating performance of recession analysis methods based on observed streamflow data

As discussed in Section 2.3, performance evaluation based on observed data could only be obtained using an indirect approach. Here the reconstructed streamflows were compared with the observed data. Following the steps presented in Fig.

2a, a total of 180 (45 watersheds × 4 recession analysis methods) streamflow time series were reconstructed based on the derived $g(Q)$ from the four methods. $g(Q)$ functions estimated by the four methods exhibit significant differences (Fig. 3), with $c_2$ estimated by *CTM* exhibiting a narrower range than that obtained from *IBM* and *EBM*. In addition, *EBM* tends to generate the largest $c_1$ and $c_2$ for the $g(Q)$ function. Significant differences among $c_1$ and $c_2$ values obtained from the four recession analysis methods highlight the role of selected methods on the derived $g(Q)$ function, and hence the *S-Q* relation.

Similar conclusion regarding the influence of recession analysis methods on the parameters of *S-Q* relation was also drawn by *Stoelzle et al.* [2013], *Chen and Krajewski* [2016], and *Dralle et al.* [2017]. Notably, the differences in $c_1$ and $c_2$ across the four methods may also impact hydrologic analysis and characterizations. This is clear from Fig. 4, which shows marked difference in the ranges of $wr^2$ for reconstructed streamflow magnitude ($wr^2(Q)$) obtained using different recession analysis methods. The median, $25^{th}$ and $75^{th}$ percentile, as well as the maximum of $wr^2(Q)$ were largest for *IBM* and smallest for

*LEM*.

     Notably, $wr^2(Q)$ was observed to be smaller than 0.5 for most of the watersheds from all four methods. Only in a few cases $wr^2(Q)$ was higher than 0.5, and all these cases used *IBM* or *EBM* for streamflow reconstructions (Fig. 4). Even for the best performing method (i.e., *IBM*), which had the largest $wr^2(Q)$ among the four methods, only 5 out of 45 watersheds displayed $wr^2(Q) > 0.5$. The result indicates that derived *S-Q* relations from either of the four methods are not guaranteed to

yield accurate streamflow using Eq. (7). As a corollary, estimation of other closely coupled water budget components such as evapotranspiration or precipitation using Eq. (7) is not guaranteed to be accurate as well. Unsatisfactory performance of the four methods for reconstructing the streamflow can be due to: (a) insufficient accuracy of the methods for capturing the true $g(Q)$ and hence the *S-Q* relation; (b) violation of assumptions inherent in Eq. (7), such as the assumption that streamflow depends solely on catchment storage, and catchment storage can be represented as a single storage element [*Kirchner*, 2009];

(c) errors in measurements of meteorological forcings and streamflows; (d) errors introduced by mapping input data across different spatial scales; and (e) the approximation involved in the calculation of *ET*. This means that evaluation of recession analysis methods based on Eq. (7) or for that matter using any other model that uses similar assumptions and approximations, may be unreliable. In light of this, here we also evaluated the performance of recession analysis methods in a control setup which ensured that aforementioned assumptions and approximations have negligible impacts.




### 3.2 Evaluating performance of recession analysis methods based on a control experiment setup

As discussed in Section 2.3, performance evaluation based on synthetic data was obtained using both direct and indirect approaches. In the direct approach, differences in estimates of $c_1$ and $c_2$ of $g(Q_{syn\_\varepsilon 1})$ (Figs. 5 (a) and (b)) were evaluated with respect to their true value, i.e., $c_1$ and $c_2$ of $g_{syn}(Q_{syn})$, for a total of 180 (45 watersheds × 4 recession analysis methods) cases.

*LEM* tended to underpredict both the intercept and slope of $g_{syn}(Q_{syn})$ function compared to other methods, while *CTM* tended to overpredict the intercept but underpredict the slope of the $g_{syn}(Q_{syn})$ function. The intercept and slope estimated by *IBM* and *EBM* were pretty close to the true value. However, error in the estimated slope by these methods can be greater than 2 for some watersheds, which indicates that the derived *S-Q* relation is significantly different from the true characteristic of the system. The result raises doubt about the accuracy of the derived *S-Q* relations in previous studies where validation of the

derived *S-Q* relation was missing. Similar evaluations were also performed for estimates of $c_1$ and $c_2$ of $g(Q_{syn\_\varepsilon 2})$ (see Section 2.3 for more details). Figs. 5 (c) and (d) again show that *LEM* underpredicted intercept and slope of the $g_{syn}(Q_{syn})$ function compared to other methods, while *CTM* overpredicted intercept and underpredicted slope. The intercept and slope estimated by *IBM* and *EBM* were again very close to the true values. These results show that across a range of noise magnitudes (or degree of spread in $\ln(-dQ/dt)$ vs. $\ln(Q)$ plot), *LEM* consistently underpredicts the intercept and slope than

the true value, while *CTM* consistently overpredicts the intercept and underpredicts the slope. Also, *IBM* and *EBM* perform better at extracting the *S-Q* relation from the noisy streamflow time series.

      Indirect evaluation of the four methods, which involved comparison of reconstructed streamflows to $Q_{syn}$, further corroborate these findings. For example, total ranks and average $wr^2$ based on indirect evaluation (see Synthetic data: Section 3.2 Level 1 noise and Level 2 noise in Table 2) showed that *IBM* and *EBM* were the top two performing methods with the

smallest total ranks. The average $wr^2$ obtained from *IBM* and *EBM* were actually very close to each other and much better than those obtained from *LEM* and *CTM*. Notably, the relative performances of the four methods showed similar trend irrespective of the reconstruction variable (e.g., $Q$, $pdfQ$, $pdfHQ$, $pdfLQ$) used for evaluation. However, this was not true when the indirect evaluation was conducted using real streamflow data. For instance, Observed data: Section 3.1 in Table 2 shows that when the relative performance of *LEM* and *EBM* was evaluated based on their total ranks in $pdfQ$, $pdfHQ$, and

*pdfLQ, LEM* performed better than *EBM*. However, when the evaluation was based on their total ranks in $Q$, *LEM* performed worse than *IBM*. This indicates that evaluation of the relative performance of different methods can get affected by errors in the input data and violation of assumptions inherent in the model (e.g., Eq. (7)) used for reconstruction of streamflow.

      It is to be noted that the variances in the estimated intercepts and slopes from all methods were smaller for level 1 noise, and the increase in the noise led to a decrease in performance for these methods as indicated by smaller interquartile

ranges and closer to zero median values in Figs. 5 (a) and (b) than in Figs. 5 (c) and (d). This reveals that fitting methods used to derive the characteristic sensitivity function, $g_{syn}(Q_{syn})$, are susceptible to noise in the data. Range of errors in the estimated intercepts and slopes by *LEM* and *CTM* methods were very close to each other and were much larger than *IBM* and *EBM* methods under low noise levels. The range became much larger for *LEM* than other methods under high noise levels.



This was partly because *LEM* used the lowest portion of $-dQ/dt$ vs. $Q$ scatter, which can be "unduly influenced by the stochastic scatter in $-dQ/dt$ when $Q$ is small" [*Kirchner*, 2009]. Even though the performance of all four methods generally worsened with increase in the standard deviation of the noise, reduction in the performance of *IBM* and *EBM* to noise was smaller than that for *LEM* and *CTM* methods. For example, 75th percentile of $wr^2(Q)$ for *IBM* and *EBM* registers negligible

decrease from level 1 to level 2 noise, while the median decreased by only around 0.05 (Fig. 6). In contrast, the 75th percentile of $wr^2(Q)$ for *LEM* and *CTM* were 0.5 and 0.8 under level 1 noise and it reduced to 0.32 and 0.43 under level 2 noise. The median of $wr^2(Q)$ also decreased by a larger amount than for *IBM* and *EBM*. This implies that *IBM* and *EBM* are relatively robust under higher noise. Notably, performances of all the methods with noisy synthetic data were much better than that in the observation data, which confirmed that the unsatisfactory reconstructions of streamflow time series were

indeed largely contributed by the approximations and assumptions inherent in Eq. (6). Cases with negligible errors in $c_1$ and $c_2$ were found to have $wr^2$ close to 1, thus confirming that indirect evaluation of *S-Q* relation based on reconstructed streamflow from Eqs. (6) and (7) can indeed be performed, as long as the inherent assumptions of the equations are satisfied in the studied watersheds.

    Overall, the results for estimation of streamflow PDFs were better than those for the flow time series (Table 2). $wr^2$

values for streamflow PDFs, $wr^2(pdfQ)$, were larger than 0.8 in 198 out of 360 simulations. In contrast, only 139 out of 360 simulations had $wr^2(Q) > 0.8$. The result indicates that streamflow PDFs can be better reconstructed using the derived *S-Q* relations. Notably, a good estimation of PDF does not guarantee a good estimation of streamflow time series as the autocorrelation structure in the PDFs has been effectively removed [*Vogel and Fennessey*, 1994]. For some watersheds, $wr^2(pdfQ)$ was acceptable even when $wr^2(Q)$ was small. For example, for level 1 noise, $wr^2(Q)$ was only 0.19 for watershed

2 but $wr^2(pdfQ)$ was around 0.75. This highlights the need to carefully interpret studies that evaluate recession analysis methods based only on their efficacy to model PDFs or FDC. Just because PDF or FDC reconstructions are good does not mean that the derived *S-Q* relation is accurate. In fact, the same can be said about evaluation based on high flows as well, as they are also well captured by all four methods (Table 2).

    Performance of recession analysis methods for reconstructing $Q$ and $pdfQ$ exhibit significant differences because

the two variables focus on different properties of the hydrologic data. Larger $wr^2(Q)$ indicate that both the timing of increase and decrease of $Q$ as well as its magnitude are well captured. In contrast, larger $wr^2(pdfQ)$ only confirms that the frequency distribution of $Q$ is well captured. For cases when a recession analysis method is not able to capture the recession rate of the hydrograph, $wr^2(Q)$ can get severely affected. However, for such cases, variations in event peaks may still get captured as they are strongly dependent on the variations in event precipitation magnitude, thus resulting in large $wr^2(pdfQ)$. This also

explains why high flows and its PDF were, in general, better captured by all four methods, than low flows (Table 2). For example, the $wr^2$ associated with high flow PDFs, $wr^2(pdfHQ)$, were larger than 0.8 in 220 out of 360 simulations. In contrast, only 16 out of the 360 simulations displayed $wr^2$ of the low flow PDFs, $wr^2(pdfLQ)$, to be higher than 0.8. Significantly poor performance of $wr^2(pdfLQ)$ is also partially due to the recurrence of zero values in the reconstructed



streamflow time series, because of overestimation of recession rates. These results highlight that high flows and its PDFs can be more accurately reconstructed than low flows, using the derived *S-Q* relations.

The evaluations based on the four reconstruction variables were performed not only using the $wr^2$, but also using other metrics such as the Nash-Sutcliffe efficiency (NSE), the logarithm Nash-Sutcliffe efficiency (logNSE), the Root Mean

Square Error (RMSE), and the coefficient of determination ($R^2$) (see Table S2 in Supplementary Materials). The relative rankings of the different methods remained almost the same, with *IBM* and *EBM* generally performing better than *LEM* and *CTM*.

**3.3 Is the relative performance of methods sensitive to alternative recession period extraction criteria?**

The last section evaluated the performance of different recession analysis methods based on standard criteria (Table 3) used

for identification of recession periods. As noted in Section 2.2, the criteria used to identify recession periods such as number of days after a precipitation event, length of continuous recession period, and streamflow magnitude lower than a threshold, vary between different methods and from one study to another. It was not known how the selection of these criteria may affect the accuracy of different methods, and if the conclusions drawn in Section 3.2 will be still valid when other recession period extraction criteria were used. Here we considered a range of values for six criteria used to identify recession periods

(Table 3). The selected range conservatively encompassed the values reported in literature. For all four recession analysis methods, the two extremes of each criteria were considered. Since three criteria are relevant to each of the four methods (Table 3), this translated to a total of $2^3$ criteria permutations for each method. So a total of 1440 (45 watersheds × 4 recession analysis methods × 8 criteria combinations for each method = 1440) streamflow reconstructions were performed for each noise level.

Results in Table 2 (see Synthetic data: Section 3.3) indicate that overall, *IBM* method performed the best across the 45 watersheds, with *EBM* method coming in second. In contrast to the previous results (in Sections 3.1 and 3.2) where *CTM* method performed significantly worse than *IBM* and *EBM* methods, here the performance of *CTM* was closer to *EBM*. If methods are ranked based on how often they feature in the top 5 in terms of their performance for a given watershed (Table 4), *IBM* and *EBM* methods again registered as the two best performing methods. These results highlight that these two

methods perform better than others more often in capturing both the variations and magnitude of streamflows. Furthermore, the results highlight the importance of identifying the best fitting scheme used to regress through ln(−dQ/dt) vs. ln(Q) scatter and the criteria used to extract the recession periods, before any further hydrological analysis. Otherwise, a high performing method such as *IBM* might yield worse result than an often low performing method such as *LEM*, if optimal criteria are not chosen. For the considered criteria ranges, the best criteria combination for each method based on the lowest total rank of

$wr^2$ over all 45 watersheds is shown in Table 3. The result shows that *IBM* and *LEM* performed the best when a larger streamflow threshold and longer interval since the precipitation event was used to extract the recession period. Selection of a larger streamflow threshold reduces the influence of noise existent at low flows. Longer interval since precipitation event





reduces the potential contaminating effect of surface flow contribution on discharge. In contrast, the best criteria set for *EBM* was obtained for a smaller streamflow threshold and a shorter interval since precipitation event. This was because *EBM* uses individual recession events to derive parameters. Very few recession events are identified when higher streamflow threshold and longer interval since precipitation event thresholds are used, thus impacting the robustness of parameters. As the

performance of *EBM* is most favorably influenced by length of time points used to identify an individual event, the criteria set that leads to this yields the best result. Performance of *LEM* was adversely impacted when the data points that were farther away from the mean ($-dQ/dt$) for a given $Q$ were used in regression fitting. It is to be noted that for some watersheds, the best performance obtained using the optimal combinations of recession analysis method and criteria thresholds may still be poor (Fig. 7). For example, for watersheds 2, 17, 37 and 43, $wr^2$ for $Q$ under level 1 noise was only 0.34, 0.30, 0.33 and

0.34 respectively, indicating that there are cases when none of the methods can capture the streamflow response time series well enough using *S-Q* relations derived from recession analysis methods.

**4 Summary and conclusions**

A total of 3510 streamflow simulations using both real and synthetic data sets were conducted to evaluate four recession analysis methods in terms of their performance to extract the *S-Q* relation and to reconstruct the streamflow time series and

its PDFs. Based on the discussion, experiment design, results, and analyses, the following may be noted:

1.  Our results corroborate earlier findings (e.g., *Basso et al.,* [2015]; *Ceola et al.,* [2010]; *Chen and Krajewski,* [2016]; *Dralle et al.,* [2017]; *Stoelzle et al.,* [2013]) that the choice of the recession analysis method and the scheme used to identify recession periods heavily affects the derived *S-Q* relation. We further show that the difference between *S-Q* relations derived from different methods are large enough to appreciably affect the streamflow response obtained

using it. The result raises a warning sign against application of recession analysis methods for watershed characterizations or flux estimations unless a thorough evaluation of the representativeness of the derived *S-Q* relation can be established for the specific watershed.

2.  Direct evaluation of representativeness of the derived *S-Q* relation in real settings is very difficult, in part due to the lack of fine temporal resolution (e.g., daily) watershed-wide storage data. While indirect approaches have been

developed to compare the ability of recession analysis schemes to estimate streamflow PDFs and FDCs, these approaches are severely affected by (a) errors in the input data, and (b) the violation of assumptions inherent in the model, such as the assumption that streamflow depends solely on catchment storage and that the storage can be represented as a single storage element. Users are encouraged to perform indirect evaluation of derived *S-Q* relations only using models that appropriately account for contributions of overland flow, evapotranspiration and

other dominant fluxes, in addition to that by groundwater storage, to streamflow. However, it is to be acknowledged that developing such a resolved but simple model is not trivial.




3.  To circumvent the challenges posed by assumptions and approximations in the model and data, here we presented a new "control setup" based experiments for evaluation of recession analysis methods. The design assumed *a priori* knowledge of storage-discharge relation in a watershed, and precipitation and evapotranspiration in it. The approach allowed both direct and indirect evaluation of recession analyses methods.Our results showed that although no one
5       method consistently performed better than the others for all watersheds, overall *IBM* and *EBM* showed better performance for obtaining the *S-Q* relation, while *LEM* performed the worst for most of the watersheds. This indicates that *IBM* and *EBM* may generally be preferred for recession analysis.

4.  As the criteria used to identify recession periods such as the number of days after a precipitation event, length of continuous recession period, and streamflow magnitude lower than a threshold, vary between different methods and
from one study to another, it is logistically infeasible to perform analyses for all possible permutations of recession period extraction criteria and regression schemes. To explore the validity of our conclusions for a range of recession period extraction criteria, we expanded our analyses to include criteria sets that encompass the values used in majority of previous studies. Although the main conclusion regarding the performance of different methods were still largely true, our results also showed that a high performing recession analysis method (e.g., *EBM* and *IBM*)
might yield a *S-Q* relation that is less accurate than a low performing method (e.g., *LEM*), if optimal criteria for recession period identification are not chosen. The study also identified the optimal criteria set for each method (Table 3). *EBM* performed the best when the minimum number of time points used to identify individual recession event was larger. Effectiveness of *IBM* was highest when a larger streamflow threshold and longer interval since the precipitation event was used to extract the recession period.

5.  Although the influence of the spread of the scatter in $-dQ/dt$ vs. $Q$ plot on parameters of *S-Q* relation has been shown [*Chen and Krajewski*, 2016], our results also show that the influences are large enough to significantly impact the accuracy of reconstructed streamflow and its PDFs. For watersheds with large noise in $Q$ and $-dQ/dt$ scatter, the derived *S-Q* relation from even the best performing methods (i.e., *IBM* and *EBM*), might still be uncertain. Notably, the degree of scatter seems to have more impact on *LEM* method than *IBM* and *EBM* methods.
As watersheds where all recession events exhibit similar decay characteristics generally have smaller scatter in $\ln(-dQ/dt)$ vs. $\ln(Q)$, in these watersheds *S-Q* relations derived from all four methods are expected to be very similar. Future studies may attempt to relate statistical characteristics of the spread in $\ln(-dQ/dt)$ vs. $\ln(Q)$ to the performance of methods, which could allow selection of the best method based on $Q$ data alone.

6.  Reconstructed streamflow time series using the derived *S-Q* relation generally captured the overall PDF of the base
streamflow, even when the temporal distribution of streamflow magnitudes was not well captured. In general, reconstructed streamflow PDFs, especially high flow PDFs are less sensitive to the noise in $Q$ and $-dQ/dt$ scatter. This is primarily because the magnitudes of high flows are strongly determined by the intensity of precipitation events. So even in cases when a recession analysis method is not able to capture the decay rate of recession



hydrograph, high flow and its frequency distribution might still be captured. In contrast, low flows are influenced by a number of factors including errors in the *S-Q* relation, high flows and evapotranspiration rates. This indicates that although a recession analysis method is able to capture the PDF of streamflow, especially the high flow PDF, might not indicate that the derived *S-Q* relation is characteristic of the watershed. This also means that evaluation of

recession analysis methods, based only on comparison of how well they can be used to reconstruct PDFs of the streamflow, especially the high flow PDFs, is not rigorous enough. The result also suggests that if the derived *S-Q* relation is to be used for estimation of streamflow PDFs, both *IBM* and *EBM* methods may be used to obtain accurate streamflow PDFs in most circumstances (Table 2).

Aforementioned conclusions should be interpreted with caution, as they are true for the recession analysis method

configurations used in this study. Future work may focus on performing the presented analysis in other hydroclimatic settings, and for a range of watershed properties, noise structures and magnitudes to further help test the applicability of aforementioned conclusions.

### Acknowledgements

This study was partially funded by National Science Foundation grants EAR-1331846 and EAR-1454983. The

meteorological time series data used in this study came from Xia et al. [2012] which is downloadable from http://ldas.gsfc.nasa.gov/nldas/.

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

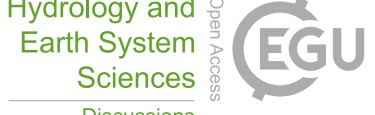

**Table 1: Watersheds used for recession analyses**

| Basin No. | USGS Gauge No. | Simulation years | Area (km$^2$) | Basin No. | USGS Gauge No. | Simulation years | Area (km$^2$) |
|---|---|---|---|---|---|---|---|
| 1 | 0209722970 | 2011,2012 | 12.10 | 24 | 02334480 | 2011,2012 | 24.19 |
| 2 | 02203863 | 2011,2012 | 22.35 | 25 | 02334578 | 2011,2014 | 13.05 |
| 3 | 02207400 | 2011,2014 | 21.11 | 26 | 02334620 | 2011,2012 | 17.87 |
| 4 | 02211375 | 2011,2012 | 10.62 | 27 | 02335350 | 2011,2012 | 23.02 |
| 5 | 02218565 | 2011,2012 | 14.71 | 28 | 02378170 | 2011,2012 | 12.90 |
| 6 | 02249007 | 2013,2015 | 9.84 | 29 | 02384540 | 2011,2014 | 21.34 |
| 7 | 02264051 | 2013,2015 | 1.79 | 30 | 02391840 | 2011,2012 | 21.57 |
| 8 | 02298492 | 2011,2014 | 15.67 | 31 | 02393377 | 2011,2014 | 9.32 |
| 9 | 02298527 | 2013,2014 | 22.56 | 32 | 02479980 | 2012,2015 | 20.93 |
| 10 | 02298530 | 2013,2014 | 17.07 | 33 | 02480002 | 2013,2014 | 21.29 |
| 11 | 02299861 | 2011,2014 | 12.72 | 34 | 03207965 | 2014,2015 | 16.06 |
| 12 | 02301738 | 2011,2013 | 7.51 | 35 | 03260100 | 2012,2014 | 10.44 |
| 13 | 02301740 | 2013,2014 | 15.77 | 36 | 03284525 | 2013,2015 | 2.49 |
| 14 | 02301745 | 2013,2014 | 5.18 | 37 | 03287590 | 2013,2014 | 10.49 |
| 15 | 02301900 | 2011,2012 | 24.60 | 38 | 03289193 | 2012,2014 | 24.79 |
| 16 | 02307668 | 2012,2013 | 9.51 | 39 | 03292474 | 2012,2014 | 15.54 |
| 17 | 02307674 | 2011,2013 | 18.16 | 40 | 03292480 | 2013,2014 | 15.02 |
| 18 | 02307780 | 2012,2014 | 3.24 | 41 | 03298135 | 2012,2014 | 14.17 |
| 19 | 02308870 | 2013,2014 | 6.50 | 42 | 03301900 | 2012,2014 | 9.06 |
| 20 | 02308935 | 2011,2012 | 6.60 | 43 | 03426470 | 2014,2015 | 19.79 |
| 21 | 02309415 | 2012,2013 | 1.48 | 44 | 03491544 | 2012,2014 | 12.10 |
| 22 | 02309421 | 2012,2013 | 8.81 | 45 | 02306500 | 2013,2014 | 19.24 |
| 23 | 02309425 | 2012,2013 | 10.59 | | | | |

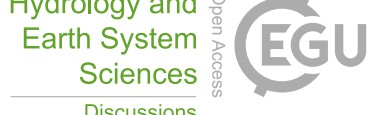


**Table 2: Total ranks and Average wr2 of all watersheds using different recession analysis methods. Better performing methods will have lower Total rank and higher Average wr2**

| | Reconstructed variables | | | | | | | |
|---|---|---|---|---|---|---|---|---|
| *Observed data: Section 3.1* | **Total ranks** | | | | **Average wr²** | | | |
| **Method** | $Q$ | pdf$Q$ | pdf$HQ$ | pdf$LQ$ | $Q$ | pdf$Q$ | pdf$HQ$ | pdf$LQ$ |
| LEM | 150 | 96 | 112 | 89 | 0.09 | 0.42 | 0.73 | 0.06 |
| CTM | 121 | 148 | 130 | 154 | 0.18 | 0.32 | 0.71 | 0.05 |
| IBM | 72 | 89 | 92 | 93 | 0.3 | 0.44 | 0.8 | 0.07 |
| EBM | 107 | 117 | 116 | 114 | 0.26 | 0.4 | 0.76 | 0.06 |
| *Synthetic data: Section 3.2* | **Total ranks (Level 1 noise)** | | | | **Average wr² (Level 1 noise)** | | | |
| **Method** | $Q$ | pdf$Q$ | pdf$HQ$ | pdf$LQ$ | $Q$ | pdf$Q$ | pdf$HQ$ | pdf$LQ$ |
| LEM | 162 | 157 | 150 | 150 | 0.3 | 0.58 | 0.7 | 0.05 |
| CTM | 144 | 147 | 146 | 114 | 0.39 | 0.66 | 0.74 | 0.18 |
| IBM | 70 | 75 | 74 | 95 | 0.82 | 0.9 | 0.9 | 0.29 |
| EBM | 74 | 71 | 80 | 91 | 0.81 | 0.89 | 0.87 | 0.31 |
| | **Total ranks (Level 2 noise)** | | | | **Average wr² (Level 2 noise)** | | | |
| **Method** | $Q$ | pdf$Q$ | pdf$HQ$ | pdf$LQ$ | $Q$ | pdf$Q$ | pdf$HQ$ | pdf$LQ$ |
| LEM | 154 | 155 | 150 | 149 | 0.17 | 0.47 | 0.63 | 0.08 |
| CTM | 156 | 150 | 147 | 111 | 0.21 | 0.51 | 0.64 | 0.17 |
| IBM | 70 | 74 | 78 | 97 | 0.72 | 0.86 | 0.87 | 0.27 |
| EBM | 70 | 71 | 75 | 93 | 0.68 | 0.82 | 0.84 | 0.31 |
| *Synthetic data: Section 3.3* | **Total ranks (Level 1 noise)** | | | | **Average wr² (Level 1 noise)** | | | |
| **Method** | $Q$ | pdf$Q$ | pdf$HQ$ | pdf$LQ$ | $Q$ | pdf$Q$ | pdf$HQ$ | pdf$LQ$ |
| LEM | 8648 | 8657 | 8209 | 7915 | 0.26 | 0.55 | 0.69 | 0.08 |
| CTM | 5849 | 6402 | 6489 | 5910 | 0.58 | 0.75 | 0.79 | 0.18 |
| IBM | 4262 | 3968 | 4002 | 4694 | 0.7 | 0.87 | 0.88 | 0.3 |
| EBM | 5001 | 4733 | 5060 | 5241 | 0.63 | 0.8 | 0.82 | 0.26 |
| | **Total ranks (Level 2 noise)** | | | | **Average wr² (Level 2 noise)** | | | |
| **Method** | $Q$ | pdf$Q$ | pdf$HQ$ | pdf$LQ$ | $Q$ | pdf$Q$ | pdf$HQ$ | pdf$LQ$ |
| LEM | 8452 | 8713 | 7919 | 7921 | 0.2 | 0.48 | 0.66 | 0.08 |
| CTM | 6721 | 6789 | 6773 | 5862 | 0.4 | 0.64 | 0.72 | 0.17 |
| IBM | 3646 | 3430 | 3781 | 4630 | 0.65 | 0.85 | 0.86 | 0.26 |
| EBM | 4941 | 4828 | 5287 | 5347 | 0.52 | 0.74 | 0.77 | 0.21 |



**Table 3: Standard criteria used in section 3.1 and 3.2, and corresponding ranges used in section 3.3 for recession period identification. The optimal criteria set for each method is identified by underlined and bolded numbers.**

| Standard criteria | LEM | CTM | IBM | EBM |
|---|---|---|---|---|
| # of days after precipitation | 5 | -- | 2 | 2 |
| # of days after which moving average of streamflow begins to decrease | -- | 3 | -- | -- |
| Maximum PET | -- | -- | 25th percentile | -- |
| Minimum # of points in each recession period | -- | 3 | -- | 5 |
| Data points lower than a fraction of standard deviation | 0.5 σ | -- | -- | -- |
| Streamflow threshold | 5th percentile | 5th percentile | 5th percentile | 5th percentile |
| **Criteria ranges** | **LEM** | **CTM** | **IBM** | **EBM** |
| # of days after precipitation | [1, **5**] | -- | [1, **5**] | [**1**, 5] |
| # of days after which moving average of streamflow begins to decrease | -- | [**1**, 5] | -- | -- |
| Maximum PET | -- | -- | [**15th**, 35th] percentile | -- |
| Minimum # of points in each recession period | -- | [2, **4**] | -- | [3, **7**] |
| Data points lower than a fraction of standard deviation | [**0.1**, 1] σ | -- | -- | -- |
| Streamflow threshold | [5th, **10th**] percentile | [5th, **10th**] percentile | [5th, **10th**] percentile | [**5th**, 10th] percentile |





**Table 4: Number of occurrences of a method's performance within top 5 for a given watershed, considered over all 45 watersheds.**

| | *Reconstructed variables* | | | | | | | |
|---|---|---|---|---|---|---|---|---|
| | *# of occurrence (Noise level 1)* | | | | *# of occurrence (Noise level 2)* | | | |
| **Method** | *Q* | pdf*Q* | pdf*HQ* | pdf*LQ* | *Q* | pdf*Q* | pdf*HQ* | pdf*LQ* |
| LEM | 18 | 10 | 22 | 20 | 11 | 14 | 19 | 30 |
| CTM | 45 | 35 | 32 | 44 | 44 | 31 | 30 | 18 |
| IBM | 88 | 99 | 86 | 99 | 95 | 108 | 102 | 110 |
| EBM | 74 | 81 | 85 | 62 | 75 | 72 | 74 | 67 |





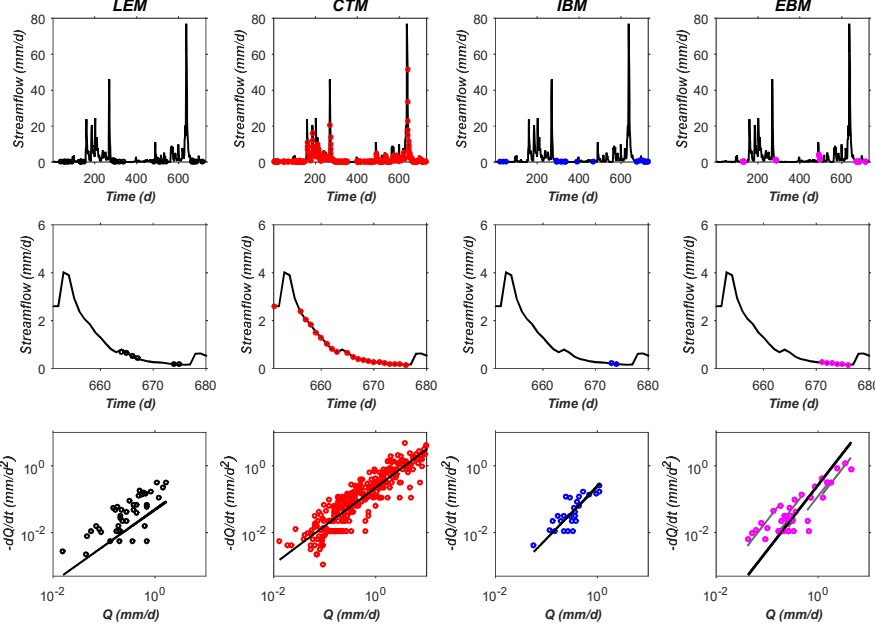

**Figure 1: Hydrograph and recession period identified by colored points using LEM, CTM, IBM and EBM methods for watershed 10 (Row 1); Zoom-in of the hydrograph and recession period shown in Row 1 from day 651 to 680 (Row 2); fitted line through (-dQ/dt) vs. Q scatter using the four recession analysis methods (Row 3).**





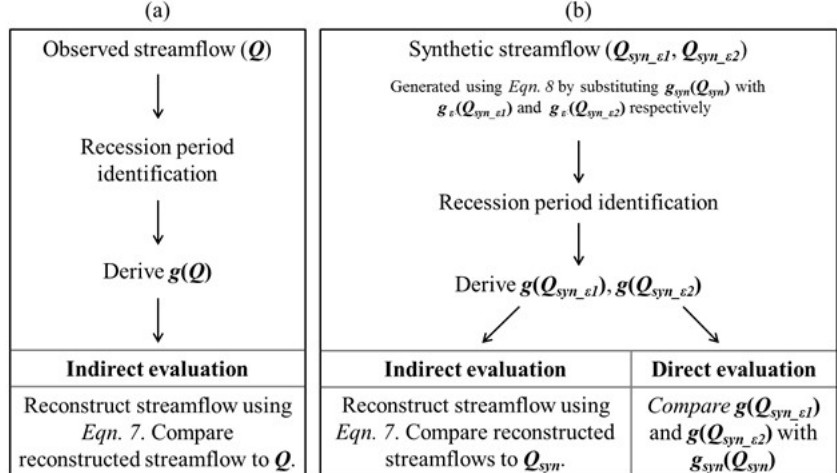

**Figure 2: Schematic flowchart of procedural steps detailed in Section 2 and implemented in Section 3**





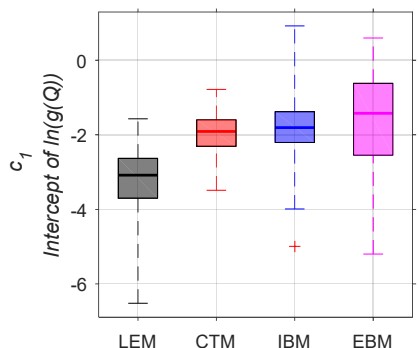 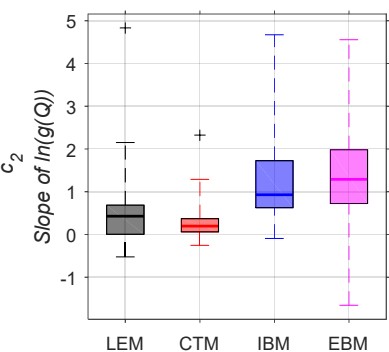

**Figure 3: Parameters $c_1$ and $c_2$ of the g(Q) function for all 45 watersheds, as estimated by the four recession analysis methods. Colored boxplots span the interquartile range, whiskers extend to three times the interquartile range. Points that lie outside this range are marked as outliers (+)**





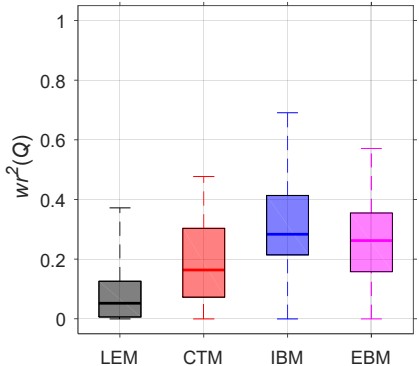

**Figure 4: Performance of the four recession analysis methods in reproducing streamflow time series (Q). Colored boxplots span the interquartile range, whiskers extend to three times the interquartile range.**





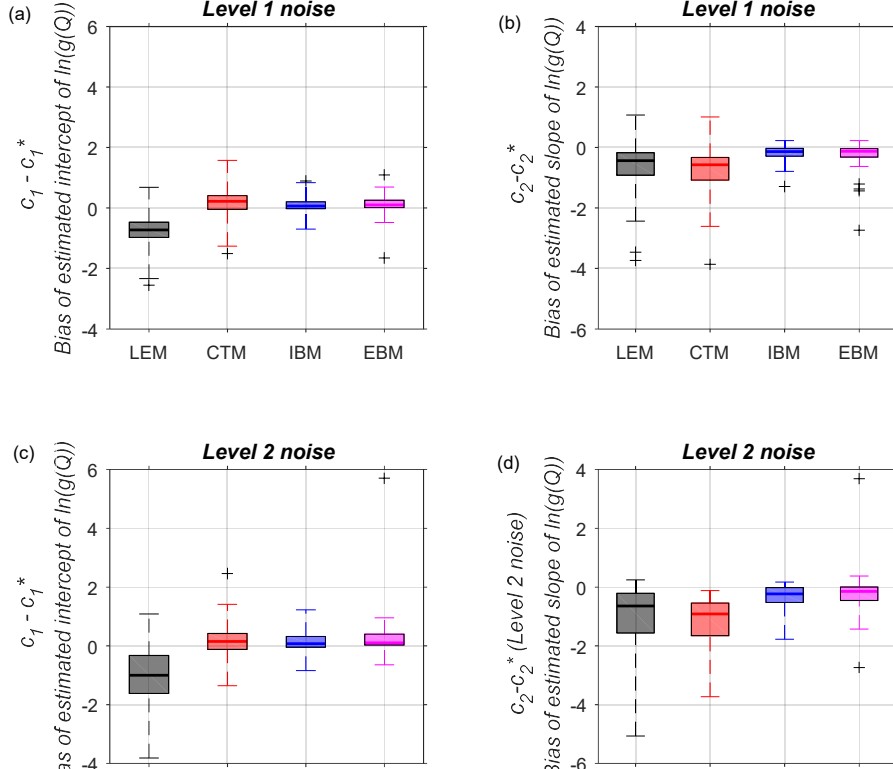

**Figure 5: Difference of estimated intercept and slope of g(Q) function from different methods with respect to the true values (identified using superscript "*") for all the 45 watersheds with level 1 ((a) and (b)) and level 2 noise ((c) and (d)). Points that lie outside this range are marked as outliers (+).**





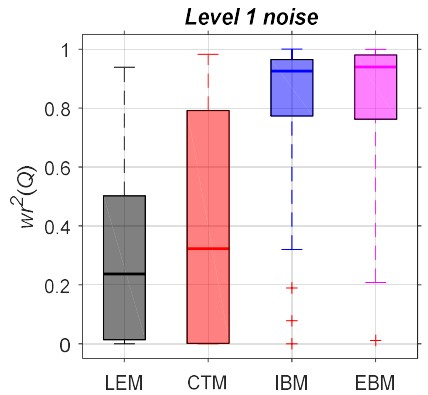 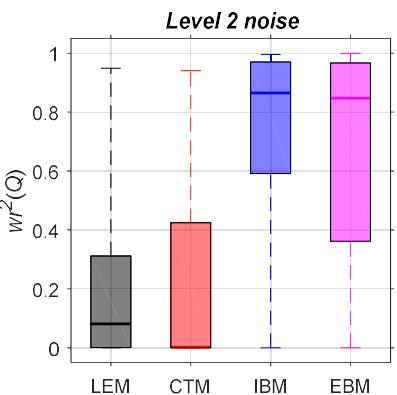

**Figure 6: Performance of the four recession analysis methods in reproducing streamflow time series (Q) in all the 45 watersheds with level 1 and level 2 noise. Colored boxplots span the interquartile range, whiskers extend to three times the interquartile range. Points that lie outside this range are marked as outliers (+).**



Figure 7: Best performing methods for all 45 watersheds and four reconstruction variables under 2 noise levels. Colors indicate the magnitude of $wr^2$ of reconstruction variables with respect to observed.