# Peer review of "On the effectiveness of recession analysis methods for capturing the characteristic storage-discharge relation: An intercomparison study"

_Hydrology and Earth System Sciences, 2018_

## Referee Comment (RC1) · Anonymous Referee #1 · 19 Mar 2018

General Remarks

The work presented in the manuscript addresses relevant problems and tries to fill important gaps of knowledge in hydrological research. The work seems to have been conducted with high rigor and is of solid quality. A systematic comparison of different storage-discharge analysis methods on a large set of catchments is an important contribution in judging the value of those methods for research and practice. The manuscript should be published after some revisions.

Streamflow reconstruction vs. "control experiment":

The reconstruction of observed streamflow is the only method presented in the paper that actually evaluates the practical applicability of the storage-discharge analysis methods for real-world problems. The "control experiment" setup allows, as the authors correctly point out, to better compare the different approaches by reducing the interference of potentially violated assumptions. However, the "control experiment" only allows to make comparative statements such as: "if a catchment can be accurately represented by a single reservoir, and discharge only depends on the state in that reservoir, then method A is superior to method B in identifying the characteristic storage-discharge relationship". It cannot be used to make statements about the applicability of those methods in general. This should be stated more clearly throughout the manuscript.

Chapter 2.2.1

Page 5, Line 20: The inequality is true only on the rising limb of the hydrograph, but we should be on the falling limb. The same for Line 21.

Chapter 2.3

Page 7, Line 23,24: How well was the lag time identifiable? Was there an uncertainty in the lag time? How would this uncertainty propagate to the streamflow reconstruction? This should be investigated and mentioned in a few sentences.

Chapter 2.4

Equation 9: The authors state that the measure of wr2 allows to account for both, the dispersion and systemic bias between two variables. They say that the slope of the regression line (b) between two variables allows to consider the systemic bias. But in my opinion, this is only partially true: the slope of a regression line alone is not sufficient to say anything about a systemic bias between the two variables. The intercept would also have to be considered for this. For example, it could be that the slope is 1, but one variable is consistently higher than the other because the intercept is not 0. Furthermore, none of the metrics in Chapter 3.2 (Page 13, Line 5) considers

a systemic bias. Therefore, it should be checked if the intercept of the regression lines significantly differs from 0 and if yes, this should be included in the analysis to make statements about the bias.

---

## Referee Comment (RC2) · C. Brauer (Referee) · 27 Mar 2018

**General comments**

In this paper 4 recession analysis methods are tested systematically for 45 catchments in the US. I think this paper is a valuable addition to the growing number of papers on recession analysis and using recession analysis results for hydrological modelling. It is mostly well-written and the research set-up is clear.

I specified some comments below (in which I abbreviate e.g. "page 1, line 2" to 1-2).

[Figure]

**Specific comments**

You used daily data. Sometimes this conflicts with the assumptions made in the recession analysis technique (e.g. assumption of ET=0). Please discuss the implications of using this resolution in more detail.

I find it difficult to assess the quality of the streamflow simulations based on $wr^2$ alone. You mention that the simulations were not good (e.g. 10-18: "Even for the best performing method (i.e., IBM), which had the largest $wr^2(Q)$ among the four methods, only 5 out of 45 watersheds displayed $wr^2(Q) > 0.5$"), but does that mean that they are reasonable, bad or terrible? Please also give some Nash-Sutcliffe efficiencies and/or Kling-Gupta efficiencies. It would also help if you would show a few typical hydrographs (for a whole year) from observations and the 4 simulations. A time series could also help to illustrate the nice analysis in 12-24..13-2. I would appreciate a bit more discussion of the simulated hydrographs. For example, did any of the catchments have zero discharge at any moment and how does the streamflow recover after dry periods (and do some recession analysis techniques lead more often to certain problems than others)?

Did the performance of the simulation correlate with the humidity (runoff ratio) of the catchments? Were the 5 catchments that performed well the wettest ones?

The results section (3.1) starts directly with the assessment of the methods by analysing streamflow. The results from the recession analysis itself are hardly described (only the first few sentences of 3.1). You do have Figure 1, which is discussed in different locations in the methods section. I think it is more clear to move all results from the recession analysis to the results section (as separate subsection). Additional analysis of the recession analysis could help to understand the differences between the obtained parameter values, and in that way also the discharge simulation. For example, it could help to show for a few typical catchments the 4 regression lines in the $(\ln(Q),\ln(-dQ/dt))$-plot, because it is the combination of c1 and c2 that determines the

streamflow dynamics.

The following paper could be useful (I always feel self-centered to recommend my own paper, but in this case I really think it could be useful, because we also compared methods to obtain parameters for the storage-discharge relation, including using storage measurements): C.C. Brauer, A.J. Teuling, P.J.J.F. Torfs, and R. Uijlenhoet (2013): Investigating storage-discharge relations in a lowland catchment using hydrograph fitting, recession analysis, and soil moisture data, Water Resources Research, 49, 4257–4264, DOI: 10.1002/wrcr.20320.

**More specific comments**

Title: maybe mention that by "capturing the characteristic S-Q-relation" you mean "to simulate streamflow", because that is how you assess the "effectiveness". You can also lead with "intercomparison" and avoid the two-part title, for example "Intercomparison of recession analysis methods for capturing the characteristic storage-discharge relation and simulating streamflow".

2-9, "These relations are often derived by identifying a relation between stream discharge, Q, and its time derivative, $-dQ/dt$, during recession periods of the hydrograph when evapotranspiration and rapid flow contributions (e.g., surface and subsurface flows) to the discharge are negligible, and streamflow is primarily determined by the catchment storage [Brutsaert and Nieber, 1977]": In the method by Brutsaert and Nieber, ET was not assumed to be zero, becasue they used daily data.

2-18, "Previous studies have used thresholds ranging from one to 10 days after rainfall events": Kirchner (2009) used several hours.

3-3 and 7-15 "However, the "true" g(Q) or the "true" S-Q relation is generally unknown for real watersheds due to lack of daily watershed-wide storage data.": Even if you do have that data, it is still difficult to get the S-Q relation 9 (as our study in 2013 showed).

[Figure]

4-17: Can catchments with a runoff ratio of 0.16 still be called humid?

5-4: Did you consider other relations than the power law, such as a polynomial?

5-6 (eq. 4): First you mainly use alpha and beta to describe the relations. Later you use c1 and c2. I think it would be better to choose one. I prefer c1 and c2 since it is easier to understand in the regression analysis and hydrograph simulation. If you choose to use both, explain right at the start (it now comes quite late, in 7-12) why you choose to use two sets of parameters and what the relation is between the two.

5-18..21: You increase delta t when delta Q is too small. This means that you get fewer data points with low Q. How does this affect the regression (more weight to high Q) and the results? Why did you choose 0.001 for the threshold? Do the results change if you change this number?

5-27: You decreased c to 0.5. Did you have enough points left to perform the analysis on? Mention how many data points you have for c=2 and c=0.5 (averaged over all catchments).

6-20, "only periods with PET less than 25th percentile were selected": This can still be large compared to the delta Q in that period. Does it become problematic in any of the catchments?

7-21: Mention (in the results) the range of the ET reduction factor lambda over the 45 catchments (add column to Table 1). Does the assumption of a constant reduction over the year become problematic in any of the catchments? I can imagine that in catchments with a low ETact/ETpot ratio, the reduction mostly occurs at the end of summer, therby influencing the hydrograph.

8-26: Why not Kling-Gupta efficiency?

10-5: Was the same period used for both recession analysis and streamflow simulation? I suppose not, but I could not find it in the text.

10-26 "the approximation involved in the calculation of ET": Do you mean the lambda factor? Please explain more clearly.

Table 1, Fig 7: How are the stations ordered? If you order them by runoff ratio, lambda or size, you could determine if there are relations in Fig 7 (what determines which method performs best).

Fig 3: You can increase the information content of this figure by combining the panels, because parameters c1 and c2 are often highly correlated. Put c1 on the x-axis and c2 on the y-axis (or the other way around). Add all 4x45 parameter sets as points, with different colors for different methods. You could even try plotting the catchment number, but that may become illegible. Along the top, draw the 4 box plots for c1 and along the right axis the box plots for c2.

**Technical comments**

1-22: "fitted a line through the lower envelope" is not completely correct, because you draw a line that surrounds the points and not one through the cloud.

4-14: all the watersheds -> all watersheds

5-9, 6-7: replace the 1 in several subscripts with delta t. For example "$t - \delta t$" instead of "$t - 1$"

5-12: envelop -> envelope

5-22: for the recession period -> for all recession periods in the whole series

5-23: in the -> with a

7-6..7: move this sentence to results section?

7-26: Eq. (7) is -> Eq. (7) are

8-12: time series was -> time series were

9-7 (eq. 9): replace the round brackets with "if"? Now it looks like a function.

9-10: B and R were -> B and R are

9-10: B was -> B is

9-11: indicate -> indicates

9-12: equal sized -> equally sized

9-20 "(Table 2)": move to results

11-7: pretty -> quite

12-9: confirmed -> suggested/indicated (because it could be due to another assumption that was not listed)

13-16: each criteria -> each criterion

13-14..13-19: move to methods section?

15-25, "where": same font size

Tab 2, caption: wr2 with superscript

Tab 2: Rotate "Observed data: Section 3.1", "Synthetic data: Section 3.2" and "Synthetic data: Section 3.3" and move them left of the rows that belong to that section. (at first I didn't understand that the 3rd and 5th block of rows belonged to the 2nd and 4th)

Tab 3: Remove duplicates: Add the numbers from the 2nd block between brackets to the 1st block (and explain in caption), Maximum PET -> Maximum ET (percentile), Streamflow threshold -> streamflow threshold (percentile), remove the words percentile from the rest of the table.

Fig 1: Add the number of points in bottom row plots as n=...

Fig 3-6: You have 2 colors and 4 methods. Either make all colors the same or all different (my preference).

Fig 5: Combine to 2 panels to facilitate comparison.: Combine boxes from left panels (so 4x2 boxes next to each other, LEM-1, LEM2, CTM1, CTM-2, etc) and the same for the rigth ones.

—

Good luck!

Claudia Brauer